# Moving Off-the-Grid: Scene-Grounded Video Representations

**Sjoerd van Steenkiste**[*,1]**, Daniel Zoran**[*,2]**, Yi Yang**[2]**, Yulia Rubanova**[2]**,**
**Rishabh Kabra**[2]**, Carl Doersch**[2]**, Dilara Gokay**[2]**, Joseph Heyward**[2]**,**
**Etienne Pot**[2]**, Klaus Greff**[2]**, Drew A. Hudson**[2]**, Thomas Albert Keck**[2]**,**
**Joao Carreira**[2]**, Alexey Dosovitskiy**[†,3]**, Mehdi S. M. Sajjadi**[2]**, Thomas Kipf**[*,2]
[1]Google Research, [2]Google DeepMind, [3]Inceptive

## Abstract

Current vision models typically maintain a fixed correspondence between their representation structure and image space. Each layer comprises a set of tokens arranged "on-the-grid," which biases patches or tokens to encode information at a specific spatio(-temporal) location. In this work we present *Moving Off-the-Grid* (MooG), a self-supervised video representation model that offers an alternative approach, allowing tokens to move "off-the-grid" to better enable them to represent scene elements consistently, even as they move across the image plane through time. By using a combination of cross-attention and positional embeddings we disentangle the representation structure and image structure. We find that a simple self-supervised objective—next frame prediction—trained on video data, results in a set of latent tokens which bind to specific scene structures and track them as they move. We demonstrate the usefulness of MooG's learned representation both qualitatively and quantitatively by training readouts on top of the learned representation on a variety of downstream tasks. We show that MooG can provide a strong foundation for different vision tasks when compared to "on-the-grid" baselines[1].

## 1 Introduction

Learning visual representations of the physical world is at the core of computer vision. Recent years have seen a surge of vision models that address this problem via self-supervised learning [5, 8, 23, 40]. By leveraging objectives such as contrastive learning [5, 8] and masked image modelling [23], great strides have been made towards learning useful representations from image data. The vast majority of these methods use convolutional networks [35], vision transformers [14, 54] or a combination thereof [4]. This choice of architecture comes to no surprise, as it inherently reflects the structure of the underlying datasets: images are (typically) represented as grids of pixels, which are conveniently and efficiently processed using 2D convolutions and patch-based heuristics. This *grid-based* processing, however, leads to an inherent entanglement between the representation structure and image structure. In other words, specific tokens or feature vectors of the representation are encouraged to capture the contents of a specific image location, instead of binding to the underlying content of the physical scene.

This issue is particularly apparent when processing video: when there is motion in the scene, either by ego-motion or object motion, the contents of the scene will move across the image plane and as such the representation (i.e. in terms of what is encoded where) will change accordingly. However,

---

[1]Project page: https://moog-paper.github.io/.
[*]Equal contribution, [†]Work done while at Google.

many down-stream scene understanding tasks require observing how individual objects (or object parts) change their configuration over time, even when other factors like camera motion translate the objects around the image plane. In this case, a representation that preserves correspondences between meaningful scene elements and representational elements is likely preferred.

As a consequence, many works targeting object-centric tasks such as object detection [4, 36], tracking [25, 33, 38], or segmentation [34], have adopted specialized architectural components that learn object-based representations: representations that are lifted from the image grid to bind to individual objects. These representations, however, are specialized to object-centric tasks and either need to be learned with detailed supervision [4, 34, 38] or have difficulty scaling to diverse real-world raw video data [19, 33].

In this paper we propose a transformer-based video model that learns representations that are "off-the-grid" (OTG) in a self-supervised manner, providing consistent features that bind to underlying scene elements, and tracking them as they move through time. Our method, *Moving Off-the-Grid* (MooG), makes extensive use of cross-attention to learn a latent set of tokens that is decoupled from the image grid: tokens are updated via cross-attention when a new input frame arrives, and decoded back into images via cross-attention. MooG can process videos of arbitrary length by iteratively updating the representation as new frames are observed.

In summary, our contributions are as follows:

- We introduce *Moving Off-the-Grid* (MooG), a novel transformer-based recurrent video representation model that is capable of learning OTG representations via a simple next-frame prediction loss.
- We qualitatively demonstrate that the OTG representation of MooG binds to different parts of the scene and tracks its content under motion, whereas a grid-based representation fails to do so.
- Finally, we demonstrate how this representation facilitates a variety of downstream vision tasks, including point tracking, monocular depth estimation, and object tracking. Our approach outperforms self-supervised grid-based baselines, such as DINO [5, 40], and performs competitively with domain-specific approaches, such as TAP-Net [12] and TAPIR [13] for point tracking.

## 2 Related Work

Transformer architectures [54] for visual tasks have gained substantial traction in the machine learning community in recent years. Starting with methods such as the self-attention architecture applied to CNN feature maps by Zambaldi et al. [62], the Image Transformer [41] and later popular approaches such as the Vision Transformer (ViT) [14], the vast majority of this class of methods operates on a *grid* of image features (e.g. patches or CNN feature maps), all the way from pixels to the final output of the transformer. This choice of representation, while extremely successful on a wide range of tasks, naturally couples representations to spatial 2D locations in image space.

The predominant approach for decoupling internal model representations from the image grid is by using *cross-attention*, where one set of tokens is updated based on the value of another set of tokens. In particular, object-centric tasks such as detection [4, 64], tracking [33, 38, 63], and instance segmentation [9, 34] have found widespread adoption of this architectural principle to learn individual object tokens that are detached from the image grid, both in supervised methods such as the Detection Transformer (DETR) [4] or GroupViT [59], and unsupervised methods such as Slot Attention [36, 57] or CLIPpy [46]. Especially when extended to multi-view observations [27, 48] and video [19, 33, 38, 65], this one-token-per-object representation allows for consistent representation of individual objects across views and frames in a video. In contrast to these approaches, our method does not assume a one-to-one mapping between OTG tokens and *objects*, but instead assigns a large set of latent tokens that can flexibly bind to any part of a scene, such as small surface elements, without committing to any particular notion of an object.

The Perceiver [28] is closely related to our work: it uses a large set of latent tokens, updated via cross-attention from visual inputs, to support a range of downstream tasks. While the original Perceiver model primarily focuses on end-to-end classification tasks, PerceiverIO [29] extends this framework to use pixel-based objectives or predict other modalities (such as audio). A single time step of our model can be seen as a variant thereof: we similarly use cross-attention to map to a latent set of tokens, and decode targets (such as pixels, point tracks, etc.) similarly using cross-attention. This type of cross-attention decoder is also used in Scene Representation Transformers [49], which

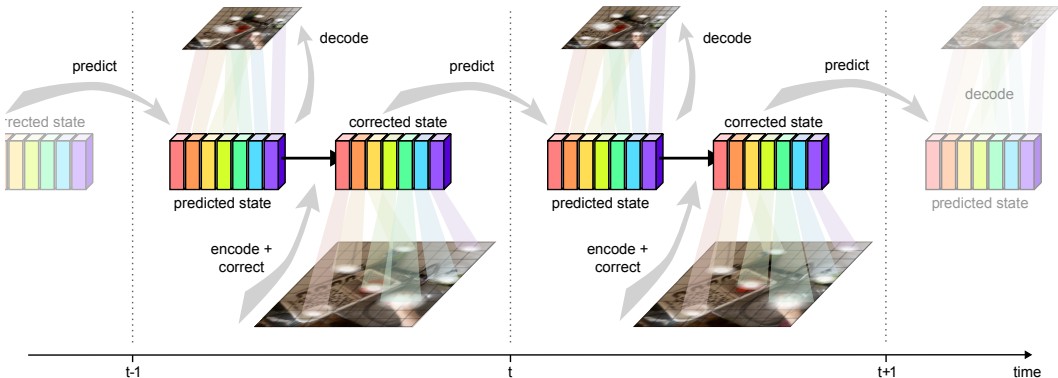

Figure 1: MooG is a recurrent, transformer-based, video representation model that can be unrolled through time. MooG learns a set of "off-the-grid" latent representation. The model first predicts a *predicted* state based on the previous model state and observation. The current observation is then encoded and cross-attended to using the predicted state as queries to produce a correction to the prediction. When training, the *predicted* state is decoded with cross-attention using pixel coordinates as queries in order to reconstruct the current frame. The corrected state is used as input to the predictor to produce the next time step prediction, and so on. The model is trained to minimize the pixel prediction error. By decoupling the latent structure from the image grid structure the model is able to learn tokens that track scene content through time.

have recently been applied to video modeling [50, 51]. Different from these works, our approach processes video in a recurrent fashion, encouraging tokens to stay consistently attached to the same scene element independently of how the camera or the content of the scene moves.

Several recent works use a separate set of tokens with either back-and-forth cross-attention, such as RIN [26] and FIT [7], or pure self-attention, with extra tokens simply appended to the original set of grid-based tokens. The latter category includes approaches such as AdaTape [61] and Register Tokens [10]. In our work, we solely update tokens by cross-attending into grid-based representations, without "writing back" into the grid-based representations.

A related line of work explores autoregressive prediction of visual tokens for self-supervised representation learning [2, 18]. Different form this line of work, MooG uses a recurrent architecture to enable consistent binding of latent tokens to scene elements (as opposed to using fixed image patches).

Naturally, most *explicit* 3D approaches for vision are "off-the-grid", such as architectures operating on top of point clouds [44, 45], and methods for 3D rendering such as 3D Gaussian Splatting [30, 37] and particle-based neural radiance fields [56, 60]. In contrast, our method does not associate explicit 3D coordinates with individual OTG tokens but instead learns high-dimensional vector representations. Outside of the field of computer vision, off-the-grid representations are the predominant representation used to model the physical world at various scales, e.g. in terms of particle-based representations for atomistic systems (one vector per atom) [20, 32, 52] or mesh-based representations for macroscopic physical systems [42], where representations are anchored to surface elements.

## 3 Method

Moving Off-the-Grid (MooG) is a self-supervised transformer model for representation learning from video. In Section 3.1 we describe its model architecture, which enables learning of scene-grounded video representations. To obtain predictions for various vision tasks, we connect readout modules to MooG's OTG representation, which we describe in Section 3.2.

### 3.1 Learning self-supervised OTG video representations

We design MooG as a recurrent model that can process an arbitrary number of video frames, while keeping a consistent scene-grounded OTG representation of the video. MooG takes as input a sequence of observed frames $\{X_t\}_{t=1}^T$, $X_t \in \mathbb{R}^{H \times W \times 3}$ and iteratively encodes them into a set of latent tokens. We separate the latent state into *corrected* states $\{z_t^c\}_{t=1}^T$, $z_t^c \in \mathbb{R}^{K \times D}$, which are obtained by encoding input frames, and *predicted* states $\{z_t^p\}_{t=1}^T$, $z_t^p \in \mathbb{R}^{K \times D}$, which are the

model's internal prediction for what will be observed at the next time step. This recurrent processing is an important part of MooG as it allows individual tokens to consistently track elements of the scene through videos of arbitrary length and "anticipate" where a scene element will be observed next.

MooG's training objective is next frame prediction given the previous frame and model state. The model is comprised of three main networks: a predictor $\mathcal{P}$ which predicts the current *predicted* state from the previous *corrected* state, a decoder $\mathcal{D}$ which decodes the *predicted* state to reconstruct the current frame and a corrector $\mathcal{C}$ which encodes the current frame and attempts to correct prediction errors made by the predictor[2]. We now describe, in order of operation, each component role and inner workings. We refer to Figure 1 for an overview of the model's structure and to Appendix C for details. For comparison, we depict a typical "on-the-grid" baseline model in Figure 2.

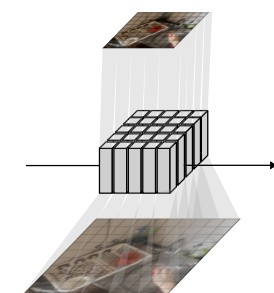

Figure 2: For comparison to MooG we here depict a classic "on-the-grid" model where tokens in the latent state are inherently tied to specific pixel locations.

**Predictor**  The predictor takes the previous *corrected* state $z_{t-1}^c$ and produces the *predicted* state for the current time step $z_t^p$:

$$z_t^p = z_{t-1}^c + \mathcal{P}(\texttt{kqv} = z_{t-1}^c). \tag{1}$$

The role of the predictor is to predict the current state based on previous observations, before the current time step's inputs are observed. The predictor network itself $\mathcal{P}$ is a simple self-attention transformer network [54]. Note that the initial corrected state $z_0^c$ is initialized to random Gaussian noise (with zero mean and $\sigma = 10^{-4}$) and is not learned. This choice of initialization comes from the need to break the symmetry among the state's tokens, as well as preventing "specialization" of tokens and maintaining permutation symmetry.

**Corrector**  The role of the corrector is to convey information from the current observation $X_t$ and use it to update the predicted state $z_t^p$ to form the *corrected* state $z_t^c$ for the current time-step. The resulting corrected state should contain the new information obtained from the observation. The image is first encoded using a convolutional neural network $\mathcal{E}$ with an added Fourier positional embedding to its output and linearly projecting the result to produce a feature grid $F_t \in \mathbb{R}^{H' \times W' \times D}$. Here $H'$ and $W'$ are the resulting spatial dimensions after accounting for striding. This feature grid is then attended to with a cross-attention transformer $\mathcal{C}$ using the current predicted state $z_t^p$ as initial queries to obtain the state update:

$$z_t^c = z_t^p + \mathcal{C}(\texttt{kv} = F_t, \texttt{q} = z_t^p), \quad \text{where} \ \ F_t = \mathcal{E}(X_t). \tag{2}$$

It is important to note that the corrected state does not receive its own individual loss and is only used to provide a better estimate for the predictor in order to predict the next step. In this sense, the separation between the corrector transformer and predictor is somewhat artificial. It is, however, crucial that the image is decoded *only* from the predicted state—decoding the current frame from the corrected state reduces the problem to simple auto-encoding and hurts representation quality considerably.

**Decoder**  The decoder takes the current predicted state $z_t^p$ and decodes it into an RGB image of arbitrary resolution. Decoding is done using cross-attention where queries are embedded pixel coordinates $P$ and keys and values come from the predicted state $z_p^t$. We utilize the same architecture for arbitrary pixel-based readouts (described in Section 3.2); see Figure 3 for a schematic depiction. Note that the states that are learned are comprised of tokens that are OTG and offer no direct correspondence to the spatial image grid, which necessitates this design of our decoder. At training time we decode $\tilde{X}_t$, a sub-sampled version of the target image for efficiency:

$$\tilde{X}_t = \mathcal{D}(\texttt{kv} = z_p^t, \texttt{q} = P). \tag{3}$$

To allow for efficient training, we decode only a randomly selected subset of pixels at each training iteration, which reduces computational demands significantly. For details see Appendix C.

Of special note are the attention weights of the decoder network $\mathcal{D}$ as they allow us to understand the relationship between a specific spatial position in the image and specific tokens in the latent representation. We analyze this relationship in Section 4.1.

---

[2]This is reminiscent of a Kalman Filter, albeit with implicitly learned dynamics and variance estimations.

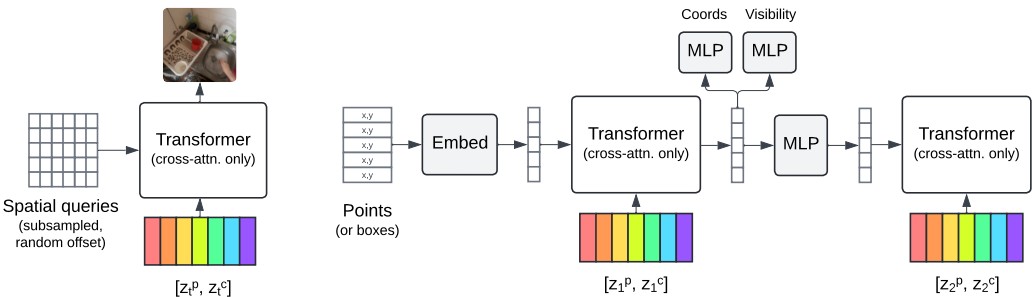

(a) Pixel readout (RGB, depth).         (b) Recurrent readout (points, boxes).

Figure 3: Readout decoders overview: for grid-based readouts (e.g. pixels), we use a simple per-frame cross-attention architecture with spatial coordinates as queries, whereas for set-based readouts (points, boxes), we adopt a recurrent readout architecture.

**Loss and training**   We use a simple $L_2$ training loss on image pixels. For each frame we decode the predicted state $z_t^p$ into a sub-sampled output image $\tilde{X}_t$, sub-sample the input image $X_t$ at the same pixel locations and calculate the per-frame loss $L_t$:

$$L_t = L_2(\tilde{X}_t, X_t). \tag{4}$$

Note that this is a next-frame prediction loss as the predicted state depends only on the previous frame and model state. During training we unroll the model over 8 frames, initializing the state with random Gaussian noise (as described above) and equally weighting all frames for the loss. The final self-supervised prediction training loss averages the $L_2$ loss across frames and pixels.

### 3.2   Readout decoders for downstream tasks

We qualitatively assess properties of the learned representation in Section 4.1. To make a quantitative assessment, we propose general readout decoders that support a variety of downstream tasks. We distinguish between two types of readouts: grid-based readouts (e.g. RGB or depth pixels) and tracking-based readouts (e.g. 2D points or object tracking). To produce the desired output, all readouts read from the OTG tokens contained in the predicted and corrected states for a given timestep. See Figure 3 for a schematic overview.

**Grid-based readouts**   For dense grid-based readouts, we reuse the same decoder architecture as we use for the pixel decoder: individual spatial locations are queried using their $(x, y)$ location in a transformer that solely uses cross-attention. For computational efficiency, we query using a subsampled spatial grid with a random offset to avoid overfitting on a particular subset of pixels.

**Recurrent, query-based readouts**   For tracking-based readouts that require keeping track of content in the video after starting from an initial location, we adopt a more sophisticated design similar to the corrector-predictor component of MooG. Given a set of $N$ queries $q_1 \in \mathbb{R}^{N \times D_q}$ (e.g. points or boxes) and a sequence of observed frames $\{X_t\}_{t=1}^T$, the task is to predict all future readout targets $\{q_t\}$ for $t = 2...T$. We associate a latent encoding $y_t \in \mathbb{R}^{N \times D_y}$ with every target at time step $t$. We first encode the queries $q_1$ using positional encoding followed by an MLP to obtain the readout latents $y_t$. Latents are processed by a corrector, followed by a predictor. Different from MooG, the corrector is implemented as a transformer which solely cross-attends into the inputs (here: MooG states $[z_t^c, z_t^p]$) without self-attention between readout latents, to avoid interaction between them. Likewise, the predictor is implemented solely by an MLP, i.e. without any self-attention between readout latents. To read out target values, we apply a learnable MLP head to the (corrected) target latents $y_t$ for each output type, eg. coords, visibility, etc.

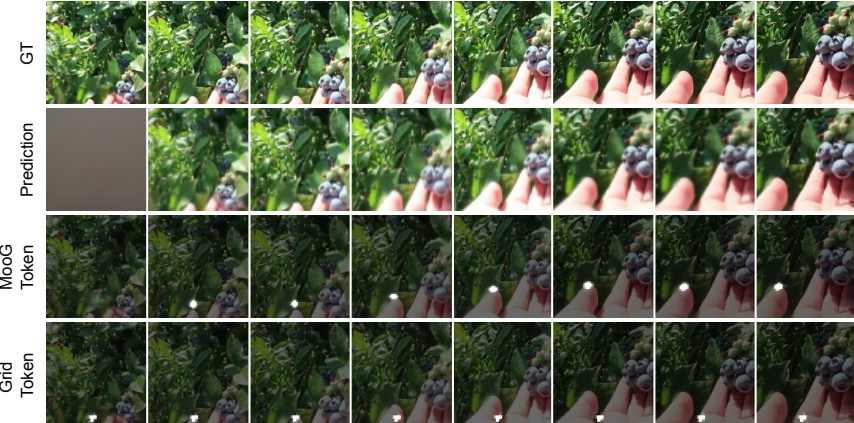

Figure 4: Qualitative analysis of MooG trained on natural videos, shown here are every 4 frames of the original 36 frame long sequence. From top to bottom: Ground truth frames, predicted frames, example MooG token attention map super-imposed on the ground truth frames, example token attention from the recurrent grid-based baseline (see text for details). As can be seen the model is able to predict the next frame well, blurring when there is fast motion or unknown elements enter the scene. The MooG attention map indicates that the visualized token tracks the scene element it binds to across the full range of motion. In contrast, the grid-based token attention map demonstrates how these tokens end up being associated with a specific image location that does not track the scene content. Please see the supplementary material (and website) for other representative examples.

## 4   Experiments

### 4.1   Self-supervised Training and Qualitative Results

We begin by qualitatively investigating properties of the learned OTG representation. We trained MooG with 1024 512-dimensional OTG tokens on natural videos from the Ego4D dataset [21] and Kinetics700 dataset [6] using only the self-supervised prediction loss in (4). The model is trained on randomly sampled sub-sequences of 8 frames, and we observe how it learns to predict next frames very well, achieving a PSNR of 25.64 (on the evaluation set) after 500K steps of training. For evaluation we unroll the model on sequences of 36 frames from a validation set — these are sequences the model has not been trained on and are much longer in duration. We first observe that the model has no trouble unrolling for much longer sequences than it was trained on, and that the predictions made are inline with the actual ground truth frames (see Figure 4). When motion is fast or erratic the model produces blurry outputs, as would be expected from a deterministic model.

**Cross-attention maps**   To understand the role of each token we focus on the cross-attention weights of the decoder, which works by having $x, y$ coordinates as queries that cross-attend into the representation in order to produce the pixel output (Section 3.1). Using the attention weights for each image location we can visualize how much each token is "responsible" for predicting a specific image location at any given time. We observe that tokens bind to the same image structure consistently through time: the attention maps of specific tokens track the image content at a particular location as it moves across the scene. A representative example of this behavior is shown in Figure 4, which we observe consistently for different tokens and on different sequences. Note that an alternative strategy for the model could have been to "tile" the image across space and make each token responsible for a specific spatial position, which is indeed is what we find the grid-based baseline tokens end up capturing (see Figure 4). For additional examples (and a better viewing experience), we refer to the videos in the supplementary material[3].

A more comprehensive way of visualizing the role of individual tokens is by taking the argmax over attention weights at each image location and visualizing the result by colour coding according to the token index. We observe the model learns to assign different parts of the scene to different tokens, tiling the image while aligning well with image structure, not unlike super-pixels. This is visualized

---

[3]Also available at https://moog-paper.github.io/.

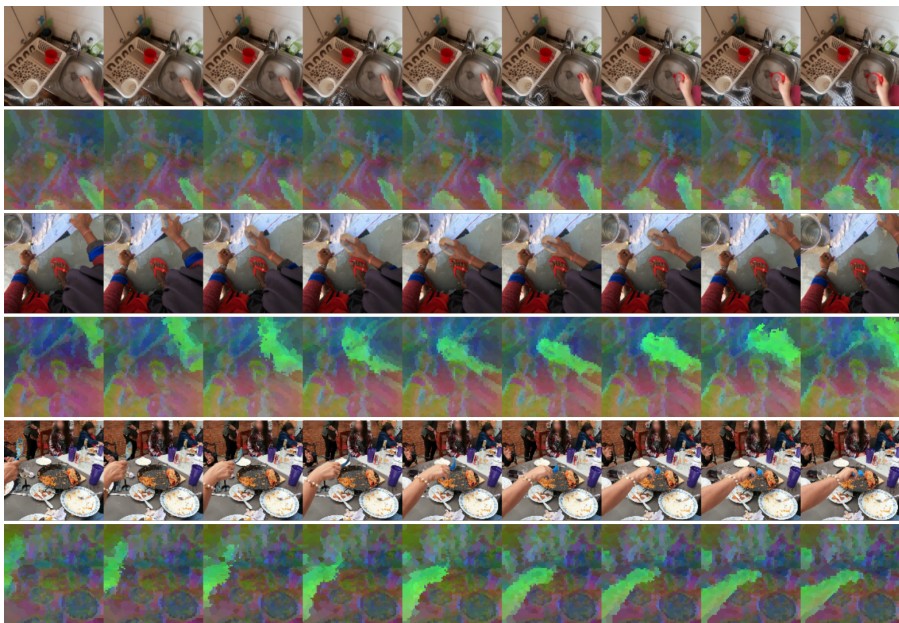

Figure 5: PCA of MooG tokens unrolled over a batch of short sequences. The model was unrolled over a batch of 24 sequences, 12 frames each. Predicted states from all time steps and batch samples were concatenated and PCA analysis was performed on the entire set jointly. We then reshape the projected set back to its original shape and use the arg-max token to visualize the result in image space (see text and Appendix for full details). Depicted are 3 of the leading PCA components in RGB. Note the salient high-level scene structure (e.g hands) learned by the model.

in Figure 7 in Appendix D, supplementary material and website. If there is good binding between specific tokens and specific image structures we should expect the colour coded image to reflect the motion present in the underlying scene. Indeed, MooG does exactly that — the model learns to devote specific tokens to specific scene structures and bind them consistently across time.

**Principal Component Analysis**    To make sense of the content of the representation at a slightly higher level we can use PCA analysis and visualize the results. We unrolled MooG on a batch of 24 sequences, 12 frames each and ran PCA on the concatenated set (across batch and time) of all tokens from the predicted state. We then project all tokens on the 64 leading PCA components and use the decoder attention weights to output 3 of the leading PCA components to image space. We observe that many of the leading components capture the positional embedding and do not relate to sequence content. However, several others end up capturing interesting (semantic) scene structure. Figure 5 shows the 3rd, 20th and 21st components in RGB space, from which it can be seen how tokens end up capturing scene content with similar projections for hands, for example, across several scenes. See website and appendix for more examples.

### 4.2    End-to-End Training and Quantitative Analysis

In the previous subsection we observed qualitatively how learned off-the-grid representations end up capturing meaningful elements of the scene. Here we study the quality of the learned representation quantitatively, focusing on three down-stream tasks: point tracking, depth prediction and object tracking. For each down-stream task we consider two different approaches for training a readout head that reflect common use cases: (1) training on top of the representations obtained from a frozen pre-trained MooG model and (2) training the readout decoder alongside MooG in an end-to-end manner, i.e. by back-propagating gradients into the model. MooG learned representations are quite local in nature due to the simplicity of the loss and the short prediction horizon. As such we do not expect it to learn abstract representations suitable for more high level tasks such as action recognition etc. We focus here on low and mid level downstream tasks. Details are available in Appendix C.

We focus on two classes of baselines in our comparison: (1) on-the-grid baselines derived from MooG, DINO [5, 40] or VideoMAE v2 [55], and (2) expert baselines that are more domain specific. The

Table 1: Down-stream readout performance from frozen representations. For DINO and VideoMAE v2 baselines we highlight what ViT configuration their encoder is based on: (S) ~20M parameters, (B) ~80M parameters, (G) ~1000M parameters. **MooG uses fewer than 35M parameters for encoder, corrector and predictor combined, which suggests a comparison to B-sized models.**

| Name | MOVi-E | | | DAVIS | Waymo |
| --- | --- | --- | --- | --- | --- |
| | Points ($\uparrow$AJ) | Depth ($\downarrow$AbsRel) | Boxes ($\uparrow$IoU) | Points ($\uparrow$AJ) | Boxes ($\uparrow$IoU) |
| MooG | **0.839** | 0.0359 | **0.793** | 0.687 | **0.730** |
| Grid | 0.769 | 0.0451 | 0.730 | 0.518 | 0.625 |
| Grid Rec. | 0.778 | 0.0443 | 0.734 | 0.559 | 0.629 |
| DINOv1 (B) | 0.518 | 0.0371 | 0.724 | 0.409 | 0.566 |
| DINOv2 (B) | 0.544 | 0.0370 | 0.738 | 0.402 | 0.559 |
| VMAEv2 (S) | 0.595 | 0.0567 | 0.700 | 0.365 | 0.567 |
| VMAEv2 (B) | 0.681 | 0.0458 | 0.736 | 0.434 | 0.611 |
| VMAEv2 (G) | 0.822 | **0.0311** | **0.793** | **0.720** | 0.708 |

former grid-based baselines have the same capacity as MooG which makes them directly comparable. We consider two variations: a simple auto-encoder with high capacity, where we have removed the corrector and predictor (named *Grid*); and a recurrent on-the-grid baseline where the corrector implements a cross-attention between the output of the encoder (i.e. on-the-grid latents) and the corrected latents from the previous step. In both cases we account for the absence of the predictor (and corrector) by adding self-attention transformer layers to the encoder. For DINO, we compute representations for each frame using the official pre-trained ViT-B/16 and ViT-B/14 checkpoints that are available for v1 [5] and v2 [40] respectively. Similarly we use the publicly available checkpoints for VideoMAE v2 [55] for three different model sizes: S/B/G. Note that the available VideoMAE v2 model size S and B are distilled from G, i.e. not trained from scratch (unlike MooG). The on-the-grid baselines (including DINO and VideoMAE v2) make use of the same readout decoders as for MooG. We discuss expert baselines in the relevant paragraphs below.

**Point tracking** The task of point tracking requires tracking of individual 2D points $(x, y)$ in a video. This can be viewed as a generalization of optical flow prediction, which similarly requires understanding of movement of physical surfaces in a scene relative to the observer. Intuitively, an OTG scene-grounded latent representation should align well with this task and support generalization.

We train MooG on Kubric MOVi-E [22] using point annotations computed in a similar manner as in Doersch et al. [12]. For each video, we randomly sample a clip of 8 frames to train on. We sample 64 points per frame and use the location of each point in the first frame as the query. To evaluate each model we report the average Jaccard (AJ) as in Doersch et al. [13], which evaluates both occlusion and position accuracy. Tables 1 and 2 report results for MooG and on-the-grid baselines. It can be seen how MooG learns representations that are better suited for this down-stream task as is evident from the considerable improvements over many of the baselines, especially in the frozen setting (Table 1).

In addition to results on MOVI-E, Tables 1 and 2 also report results for a zero-shot transfer setting to the real-world DAVIS dataset [43] using point tracks from Doersch et al. [12]. Interestingly, it can be seen how in the end-to-end setting (Table 2) the gap between the grid-based models and MooG decreases considerably, suggesting that while off-the-grid representations generalize better in the pre-training scenario, when optimized for a specific downstream task on-the-grid representations can still be competetive. We also compare to two expert baselines—TAP-Net [12] and TAPIR [13]—which are designed specifically for point tracking and achieved state-of-the-art performance when published. Both models use explicit cost volumes, i.e., exhaustive comparisons of a query point's features with features on every other frame, and have steps which detect peaks in the similarity maps. Our model is more general and does not have explicit mechanisms such as these at the representation level. The readout mechanism is also quite general. All of these steps exploit the nature of point tracking as a one-to-one feature matching problem, making it difficult to adapt the architecture to other problems. In Table 3 we directly compare MooG (trained end-to-end using a slightly larger encoder backbone) to these methods and report results for 8 frames as well as for the full sequence length. It can be seen how on 8 frames, MooG rivals TAPIR's performance, despite being a less specialized approach.

Table 2: Down-stream readout performance trained in an end-to-end manner.

| Name | MOVi-E | | | Davis | Waymo |
| | Points (↑AJ) | Depth (↓AbsRel) | Boxes (↑IoU) | Points (↑AJ) | Boxes (↑IoU) |
|---|---|---|---|---|---|
| MooG | 0.886 | 0.0263 | 0.803 | 0.778 | **0.719** |
| Grid | 0.860 | 0.0264 | 0.775 | 0.644 | 0.615 |
| Grid Rec. | **0.902** | **0.0233** | **0.806** | **0.779** | 0.675 |
| DINOv1 | 0.698 | 0.0381 | 0.728 | 0.578 | 0.557 |
| DINOv2 | 0.732 | 0.0439 | 0.734 | 0.656 | 0.607 |

**Monocular depth estimation**  Monocular depth estimation is a well-studied computer vision task that requires estimating the distance of surface elements in the scene from the camera. To test whether a scene-grounded representation, i.e. a representation that consistently tracks surface elements in the scene, facilitates the task of depth estimation, we train a depth readout module.

We train MooG using the depth annotations available in Kubric MOVi-E [22] normalized using $\log(1 + x)$. Similar to before, we randomly sample a clip of 8 frames to train on for each video. As our evaluation metric we report the mean of the absolute relative error (AbsRel), which is a standard metric in the literature [17]. Tables 1 and 2 report results for MooG and the grid-based baselines. Though we observe considerable improvements from the representations learned by MooG in the frozen setting, the *Grid* baseline performs comparable in the end-to-end case. This isn't surprising given that monocular depth estimation is a dense prediction task that can be learned well with on-the-grid representations. However, the results in Table 1 highlight how, when learning general representations that are not specific for a single task, the representations learned by MooG are still favorable — similar to DINO and VideoMAE v2 representations for this task when considering similar model sizes (VideoMAE G, having 1B parameters and having been pre-trained on large scale data, performs slightly better than our much smaller model on some tasks).

As an alternative baseline, we compare to DPT [47] trained on the Waymo Open dataset [53], starting from pre-trained ViT models as in Dehghani et al. [11]. All models are trained end-to-end. Table 4 demonstrates how MooG is able to outperform these on-the-grid baselines in this setup.

**Object tracking**  Tracking individual objects in a video not only requires spatio-temporal tracking of surface elements (as in point tracking), but also a broader semantic understanding of "objectness" to disambiguate object boundaries from surrounding scene elements.

We train MooG using the box annotations available in Kubric MOVi-E [22]. For each video, we randomly sample a clip of 8 frames to train on and we make use of all available boxes. As queries we use the location of each box in the first frame, and we report the average IoU (excluding the first frame in the sequence for which GT is provided) as in prior work [19, 33]. Tables 1 and 2 report results for MooG and the grid-based baselines. Similar to the point tracking results, it can be seen how overall MooG performs considerably better for object tracking compared to the on-the-grid baselines. Notice how, unlike for points, the box annotations focus specifically on objects (excl. background).

As an 'out-of-distribution' evaluation we also reports results on the Waymo Open [53] dataset, where we evaluate the model (and readout decoder) without additional training. Here we observe that the representations learned by MooG are better suited for this task in both settings. We relate the performance of MooG to a fully end-to-end supervised SAVi++ [19] model, by training and evaluating MooG on Waymo directly in the same set-up for 250K steps. In this set-up MooG achieves 0.667 IoU compared to a fully-supervised end-to-end trained SAVi++ variant that is reported to achieve 0.676 IoU in Elsayed et al. [19], despite only the readout decoder being supervised in MooG.

## 4.3 Analysis

**Number of readout layers**  The default hyperparameters for our readout decoders are designed to be sufficiently expressive to learn a general mapping between latent representations and targets (e.g. point tracks). In particular, we purposely use multiple transformer layers, as is common in the literature [29, 49]. To determine to what extent the readout decoder capacity affects our results, we repeat our end-to-end point tracking experiment, but using a single layer point readout decoder.

Table 3: Points comparison. End to end.

| Name | Davis-8 (↑AJ) | Davis-full (↑AJ) |
|---|---|---|
| MooG | **0.824** | 0.510 |
| TAP-Net | 0.687 | 0.392 |
| TAPIR | **0.823** | **0.580** |

Table 4: Depth comparison. End to end.

| Name | Waymo (↓AbsRel) |
|---|---|
| MooG | **0.094** |
| DPT (ViT-L/16) | 0.161 |
| DPT (ViT-E/14) | 0.158 |
| DPT (ViT-22b) | 0.154 |

Figure 6 shows how using only a single layer yields a marginal improvement on MOVi-E, while on DAVIS-8 a slight drop in performance can be observed.

**Number of tokens**    An advantage of having an "off-the-grid" representation is that the number of tokens is independent of input frame resolution. Furthermore, because we initialize the representation randomly, and because none of our model parameters depend on the number of tokens in the representation, we can instantiate the model with a different number of tokens at test time. Indeed, in Figure 7 we qualitatively show how MooG adapts to this change elegantly and is is still able to predict future frames well even with half or quarter of the number of tokens used in training. The model makes tokens cover larger areas of the scene to adapt for this change (these results are best view in video format). Quantitatively in Figure 6 it can be seen how increasing the number of tokens during training to 2048 or decreasing to 512 doesn't significantly affect MooG's performance.

## 5   Conclusion

While the vast majority of computer vision advances in the past decade can be attributed to successful "on-the-grid" architectures such as CNNs and Vision Transformers, the physical world ultimately does not live on a pixel grid. Instead of coupling the visual processing architecture to the architecture of the camera sensor (a pixel grid), we here propose to move visual representations off the image grid. Our MooG architecture allows representations to flexibly bind to scene surface elements and track the content of the scene as it is subject to motion. We demonstrated that this representation can serve as an effective alternative to the established grid-based counterpart, facilitating tasks that require understanding of motion and scene geometry. The proposed model in this paper is still quite simple - it is deterministic and ignores uncertainty inherent to the prediction task, and uses a very simple L2 pixel loss as the objective. A possible next step is to introduce stochasticity into the model to account for this inherent uncertainty, improving the representations and allowing for longer term prediction. The latter may help the model learn richer, higher-level features of the scene. We have observed that MooG struggles when applied to more semantic downstream tasks and this can likely be explained by the simple deterministic nature of the prediction task and the short-term prediction horizon of the model.

## Acknowledgments and Disclosure of Funding

We would like to thank Lucas Beyer and Mathilde Caron for making their internal implementation of DINOv1 [5] and DINOv2 [40] available for comparison. Similarly, we would like to thank Xinchen Yan for making VideoMAEv2 [55] available internally.

This work was carried out at Google.

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

## A   Limitations

Despite MooG's simple and scalable design and our quantitative improvements over the on-the-grid baselines, there are a number of limitations and open problems worth mentioning here.

The evaluations we have used focused primarily on readout tasks that require tracking scene content (like objects or points) or capturing its geometry (like depth prediction). In contrast, there are many other possible downstream tasks we might want a video representation model to support, including semantic segmentation, classification, or generation. Though in Figure 5 we have seen some preliminary evidence that MooG learns about structure in the scene that captures semantics, it is unclear to what degree these kinds of readouts are well-supported by OTG representations and compare to on-the-grid alternatives.

Another potential limitation of OTG representations is the behavior of the tokens when scene content disappears or (re)appears. Indeed, it is reasonable to assume that a grid-based representation changes more gradually along the boundaries and provides a clearer expectation to the decoder in terms of what content is encoded where. In contrast, an OTG representation may latch onto entirely new scene elements as they (re)appear. This is a general limitation of OTG representations, including prior approaches for learning slot-based object-representations [19, 33].

Finally, we have not investigated the scaling behavior of MooG in-depth, both in terms of scaling the size of the model as well as the amount of pretraining data.

## B   Broader Impact Statement

The focus of our work is on training more capable video models for representation learning. These representations can be used for improving on down-stream tasks, as we have done for point tracking, depth estimation, and object tracking. There are many applications that could benefit from such improved capabilities, including in the domain of robotics and classic computer vision applications. As for many computer vision systems, these improvements may also transfer to applications with negative societal impact such as surveillance.

## C   Experiment Details

### C.1   Datasets

**MOVi-E**   The MOVi-E dataset is part of the MOVi benchmark that was introduced with the release of Kubric [22], which is available under an Apache 2.0 license[4]. The MOVi-E dataset makes use of 380 high-resolution HDR photos as backgrounds and 1028 3D-scanned everyday objects [15]. Each scene consists of 10–20 static objects and between 1–3 dynamic objects that are tossed into the scene.

---

[4]https://github.com/google-research/kubric/blob/main/LICENSE

The camera moves on a straight line with random constant velocity, whose starting point is sampled randomly in a half-sphere shell around the scene. The camera is always pointed towards the origin of the scene. The training set contains 97500 videos and the validation sets 250 videos, each of length 24.

**Waymo Open Dataset**    The Waymo Open dataset [53] contains high-resolution videos ($1280\times1920$ resolution). We follow the approach the same approach as in prior work [19] for processing the data, where we only use the video recorded from the front of the car, down-sampled to $128 \times 192$ resolution. The original dataset we use contains 798 training and 202 validation scenes that span 20 seconds each, sampled at 10fps. The Waymo Open dataset is licensed under the Waymo Dataset License Agreement for Non-Commercial Use (August 2019)[5].

**Kinetics700 v2020**    The Kinetics dataset contains about 545,000 video clips of 250 frames at 25fps each. The videos depict people performing a large variety of actions in diverse environments. We temporally sub-sample the clips randomly to the desired sequence length in training (8 frames in all experiments here). The videos are randomly cropped spatially and scaled to $128 \times 128$ pixels. Evaluation is done on the validation set which contains about 34,000 clips. The dataset was released under the Creative Commons license.

**Ego4D**    Ego4D is a dataset of ego-centric videos. It has 24,000 long videos at 24fps which were sub-sampled into 2.4M 128 frames long clips. Of which further sub-sample 8 frames long clips during training. The videos are randomly cropped and down-sampled to $128 \times 128$ pixels. No other augmentation is performed. The Ego4D dataset is used in accordance to the original dataset license[6].

### C.2    Training & Evaluation

**Training**    For our quantitative evaluation we primarily train MooG and grid-based baselines on videos from Kubric MOVi-E [22]. During training, we replicate each video 3 times to reduce bandwidth when data loading. For each video, we randomly sample a clip of 8 frames to train on, and apply random crop and color augmentations. For random crops, we ensure that crops span an area inbetween $0.3$ and $2.0$ times the starting image resolution, and have and aspect ratio that lies inbetween $0.5$ and $2.0$ times the starting resolution. After applying random crop augmentations to videos of $256 \times 256$ resolution, we resize the resulting crops to $128 \times 128$ resolution. For color augmentations, we randomly decide to adjust the video brightness (up to a maximum of $32/255$ relative change), saturation (between 0.6 and 1.4), contrast (between 0.6 and 1.4 times) and hue (up to a maximum relative change of 0.2) of the video with $p = 0.8$, followed by a $p = 0.2$ chance of converting the video to grayscale.

On Waymo Open, we train on subsampled sequences of 16 frames using Inception-style random crop augmentations, where we ensure that at least 75% of the original frame is covered, before resizing back to $192 \times 128$, followed by a central crop to $128 \times 128$.

**Point Tracking**    On MOVi-E, we use point annotations computed in a similar manner as in Doersch et al. [12]. We sample 64 points per frame from a random grid having stride 4, while ensuring that at most $10\%$ of the points cover a single object. We use the location of each point in the first frame as the query, and mask out points that are occluded throughout the entire sequence or occluded in the first frame (such that no query can be provided to the decoder). To evaluate each model we report the average Jaccard (AJ) as in Doersch et al. [13], which evaluates both occlusion and position accuracy.

For evaluation on DAVIS [43] we use "TAP-Vid-Davis" labeled in Doersch et al. [12] for the DAVIS Validation set, consisting of 30 videos. We downsample to $128 \times 128$ resolution. Since MooG is an auto-regressive architecture we adjust the query points to correspond to the location of the target points in the first frame. We evaluate TAP-Net and TAPIR on the same set of points for a fair comparison.

**Monocular Depth Estimation**    On MOVi-E, we use the depth annotations that are readily available for Kubric MOVi-E [22]. Following prior work [19], we transform depth values using $\log(1 + d)$, where d is the distance of a pixel to the camera. As our evaluation metric we report the mean of

---

[5]https://waymo.com/open/terms.
[6]https://ego4d-data.org/pdfs/Ego4D-Licenses-Draft.pdf

the absolute relative error (AbsRel), which is a standard metric in the monocular depth estimation literature [17]. We subsample the targets by a factor of 4 for evaluation.

On Waymo Open, we follow the procedure outline in Elsayed et al. [19] to obtain sparse depth values from LiDAR. Points in image space for which no depth signal is available are masked out in the metric and loss. For the DPT baselines, we sample random frames from the available training and evaluation videos for train and eval respectively. For evaluation we directly central crop to $128 \times 128$, and evaluate on 10 frames at the original resolution.

**Object Tracking**   On MOVi-E we train MooG using the box annotations available in Kubric MOVi-E [22]. As queries we use the location of each box in the first frame (represented as $y_{min}, x_{min}, y_{max}, x_{max}$), and we report the average IoU across the sequence (excluding the first frame in the sequence for which ground-truth is provided) as in prior work [19, 33].

On the Waymo Open dataset [53] we evaluate on sequences of 8 frames in the zero-shot transfer setting to compare to grid-based baselines. To compare to SAVI++, we train on Waymo open at resolution $192 \times 128$ following the procedure outlined in Elsayed et al. [19] with random augmentations for sequences of 16 frames, and evaluate on sequences 10 frames. In both settings, we discard bounding boxes that cover an area of $0.5\%$ during training an evaluation, and keep a maximum of 10 boxes during training and for evaluation.

### C.3   Model Details

#### C.3.1   MooG

**Network Architecture**   The architecture of MooG for self-supervised training on video is divided into four components: *encoder*, *corrector*, *predictor*, and *decoder*, totalling approximately 35M parameters. To encode each frame, we use a convolution encoder as outlined in Table 5, followed by a Fourier positional encoding using 20 Fourier bases, which we add to the encoder output features using a single dense layer as a projection. For comparing to domain-specific baselines (TAPIR, DPT, SAVi++, etc.) we also explore a slightly stronger backbone, where we omit the striding in the last layer of the CNN and concatenate the positional encoding (as opposed to first projecting and then adding).

At timestep 0, we initialize the latent representation having 1024 tokens of size 512 by drawing from a standard normal multivariate Gaussian distribution scaled by a factor of 1e-4. The predictor predicts the state of the tokens for the next time step, which uses a 3 layer self-attention transformer as outlined in Table 7. The corrector updates the prediction based on the encoded observation, which is implemented as a 2 layer transformer that uses both self-attention and cross-attention (Table 7), where queries are computed from the predicted tokens, and the key and values are computed from the encoded observation ($32 \times 32$ patches). We apply Layer Normalization [1] to the output of the corrector, which we found to be important for long-sequence rollouts. To decode the back to pixel space, we use a 6 layer cross-attention transformer as outlined in Table 7. Queries are computed from the coordinate grid (after concatenating a Fourier positional encoding with 16 bases), and keys and values from the predicted tokens.

For the down-stream readouts we make use of the same cross-attention Transformer backbone for each readout as seen in Table 6. For spatial readouts (like depth), we follow the design of the pixel decoder where queries are computed from the coordinate grid (after concatenating a Fourier positional encoding with 16 bases), and keys and values from the tokens (here using both the predicted *and* the corrected tokens). For tracking based tasks (like points and boxes prediction), we associate a 512-dimensional latent state with each track that is initialized from the first-frame queries using a Fourier positional encoding as before, followed by a two-layer MLP to project to the desired dimensionality. The transformer backbone in Table 6 is used to update these latent states at each step by cross-attending into the predicted and corrected tokens, using the latent states as queries. A per-latent predictor (implemented by a 2-layer MLP) acts as a predictor to initialize the tokens for subsequent steps.

For our transformer implementation, we mostly follow the standard design in Vaswani et al. [54], but using the pre-LN configuration as described in Xiong et al. [58]. We also include a few recent improvements based on Dehghani et al. [11]. In particular, we apply RMS norm to the queries and keys before computing the attention weights, and execute the cross- and self-attentions paths

Table 5: Encoder CNN.

| Features | Kernel | stride |
|---|---|---|
| 64 | $3 \times 3$ | $1 \times 1$ |
| 128 | $3 \times 3$ | $1 \times 1$ |
| 128 | $3 \times 3$ | $1 \times 1$ |
| 256 | $3 \times 3$ | $2 \times 2$ |
| 256 | $3 \times 3$ | $1 \times 1$ |
| 512 | $3 \times 3$ | $2 \times 2$ |

Table 6: Readout Transformer Backbone. XA: Cross-attention. SA: Self-Attention.

| Type | Type | Layers | QKV size | Heads | MLP size |
|---|---|---|---|---|---|
| Points | XA | 3 | $64 \times 8$ | 8 | 2048 |
| Depth | XA | 3 | $64 \times 8$ | 8 | 2048 |
| Boxes | XA | 3 | $64 \times 8$ | 8 | 2048 |

Table 7: Transformer Layers. XA: Cross-attention. SA: Self-Attention.

| Component | Type | Layers | QKV size | Heads | MLP size |
|---|---|---|---|---|---|
| Corrector | XA & SA | 2 | $64 \times 8$ | 8 | 2048 |
| Predictor | SA | 3 | $64 \times 4$ | 4 | 2048 |
| Decoder | XA | 6 | $64 \times 2$ | 2 | 2048 |

in parallel (where applicable), but not the mlp path. Finally we apply another layer of Layer Normalization [1] to the output of the transformer.

**Subsampled decoding**  Instead of decoding the full image at each time-step during training we sub-sample the coordinate grid and embed the result. We first generate a sub-sampled grid of pixel coordinates $G \in \mathbb{N}^{h \times w \times 2}$ where $h = H/S, w = W/S$ and $S$ is the sub-sampling factor. In order to prevent over-fitting to specific coordinate grids we randomly offset $G$ by a random integer between 0 and $S$. We use a Fourier positional embedding to embed $G$ into a flattend embedding $P \in \mathbb{R}^{(hw) \times C}$ and use the result as the initial queries in the multi-layer cross-attention transformer $\mathcal{D}$, the output of which is projected to 3 dimensions and reshaped back to image size to form the decoded frame $\tilde{X}_t \in \mathbb{R}^{h \times w \times 3}$: Using a sub-sampling factor $S > 1$ allows the model to decode only a subset of the pixels in each frame reducing computational demands significantly.

**Training**  We train MooG on raw video data (see below for datasets used) for 1M steps using Adam with Nesterov momentum [16, 31] using a cosine decay schedule that includes a linear warm-up for 1000 steps, a peak value of 1e-4 and an end value of 1e-7. Updates are clipped using a maximum global norm of 1.0, and we use $\beta_1 = 0.9, \beta_2 = 0.95$ inside Adam. We use a batch size of 128 for most of our experiments, and a batch size of 256 for the comparison to domain-specific baselines in Tables 3 & 4. On the Waymo Open dataset we train for 500K steps for end-to-end depth prediction and for 250K steps for end-to-end box prediction to compare to SAVi++. Our MooG runs make use of 64 TPUv3 [39] chips having 32GiB memory, which each take about 48 hours for 1M steps. We implemented MooG in JAX [3] using Flax [24].

Our main training loss is a simple L2 reconstruction loss, which we compute for a subset of the pixels inspired by Sajjadi et al. [49]. Down-sampling is implemented via striding using a factor of 8 and having random offset during training. For depth readouts we use a masked L2 loss (based on the availability of the depth signal at a given location) using the same down-sampling approach. Similarly, for box readouts we use an L2 loss between the prediction and the normalized box coordinates. Finally, for point readouts, in addition to predicting their values (normalized image coordinates), we also predict whether they are visible and a measure of certainty of the prediction; for the purposes of evaluation, we only output a point as visible if the model predicts (having confidence over 50%) that it is both visible and certain of the position. This set-up is identical to the one used in Doersch et al. [13] and we adopt the same combination of Huber and Sigmoid Binary Cross Entropy losses for these terms. We amplify the Huber loss that measures the prediction accuracy by a factor of 1000 (note coordinates are normalized to between 0 and 1) relative to the visibility and uncertainty losses. For points that have left the scene we only use the visibility part of the loss.

### C.3.2 Grid-based baselines

**Grid (Rec.)** We use the same approach to training and evaluating the grid-based baselines, which only differ in their network configuration. In particular, the *Grid* baseline uses the same encoder described in Table 5 and decoder and readouts described in Tables 6 & 7. The key difference is that it does not include the corrector or predictor from Table 7, but treats the output of the encoder as the representation. To make up for the lost parameter count, we augment the encoder with a self-attention Transformer having 3 layers, QKV size of $64 \times 8$, 8 heads, an MLP size of 2018 and a hidden size of dimensionality 512.

The grid-based baseline with recurrence (*Grid Recur.*) also uses the same encoder described in Table 5 and decoder and readouts described in Tables 6 & 7. It does not make use of the predictor, but re-purposes the corrector as such. In particular, tokens for time-step $t$ are initialized from the output of the encoder (yielding a similar amount of 1024 tokens of dimensionality 512) in an on-the-grid manner. The corrector uses these tokens as queries to cross-attend into the corrected tokens from the *previous* timestep, which implements the recurrence. To make up for the lost parameter count, we augment the encoder with a self-attention Transformer having 3 layers, QKV size of $64 \times 8$, 8 heads, an MLP size of 2018 and a hidden size of dimensionality 512. The decoder reads from the output of the encoder (the tokens initialized on-the-grid) as is the case for the *Grid* baseline, while the other readout modules have access to the corrected tokes as well (similar to in MooG).

**DINO** To evaluate on DINO [5, 40] representations we make use of the official pretrained checkpoints available online[7]. We use the base ViT size, which exceeds MooG in the number of parameters. To evaluate on videos, we compute DINO features for each frame (after resizing to $224 \times 224$ and normalizing pixels the ImageNet value range) that are fed into the same readout decoders outlined in Table 6. We do not backpropgate the task-loss into the encoder in the frozen setting, while in the end-to-end setting we do.

**VideoMAE v2** To evaluate on VideoMAE v2 representations we consider 3 publicly available checkpoints[8]: the ViT-Small (`vit_s_k710_dl_from_giant`) and ViT-base (`vit_b_k710_dl_from_giant`) variants, which contain 22M and 83M parameters respectively, as well as a ViT-giant variant (`vit_g_hybrid_pt_1200e_k710_ft`) containing 1B params. The smaller variants were obtained by distilling the predictions of the ViT-giant model. We note that MooG contains approximately 35M parameters, which includes the pixel decoder. Another important difference to highlight is that the ViT-giant teacher network was finetuned for action recognition, while the ViT-small and ViT-base models were initialized from finetuned networks as well. To evaluate on videos, we first resize the video spatially to $224 \times 224$, after which we apply the VideoMAE v2 encoder to obtain a feature representation. Next, we upsample the feature representation temporally by a factor of two to recover the original sequence length. The resulting representations are fed into the same readout decoders outlined in Table 6. We do not backpropgate the task-loss into the encoder.

### C.3.3 Domain-specific Baselines

**TAP-net** For TAP-Net, we use the default released model from TAP-Vid paper [12], which encodes every frame (including query frame) at 256x256 resolution, runs through a TSM-ResNet backbone (with time shifting) and produces a $32 \times 32$ feature map. The feature map then is used to compute the correlation volume and output predict location and occlusion flag. No uncertainty estimation is used here. The TAP-Net is trained on the Kubric MOVI-E dataset using 24 frames and 256 points per frame. There is no temporal processing in the model. Every frame is treated independently and only correlation volume is used for prediction.

**TAPIR** For TAPIR, we mainly use the online model variant for comparison due to the autoregressive nature of MooG and using query points that are always sampled from the first frame, again using the publicly-released model. The online TAPIR model use causal convolutions that only receive context feature from history for updating the current frame prediction during iterative refinement. For the

---

[7]Checkpoints can be downloaded from https://github.com/facebookresearch/dino and https://github.com/facebookresearch/dinov2 respectively.

[8]Checkpoints can be downloaded by following the instructions at https://github.com/OpenGVLab/VideoMAEv2/blob/master/docs/MODEL_ZOO.md

backbone, it uses 18-layer ResNet (with strides 1, 2, 2, 1 and instance normalization) instead of TSM-ResNet, which is why it is frame independent but yields comparable performance. The backbone encodes $256 \times 256$ resolution and output two feature maps at $32 \times 32$ and $64 \times 64$ resolutions. The $32 \times 32$ feature map is used for global correlation volume same as TAP-Net, then the $64 \times 64$ is used with $32 \times 32$ together for iterative refinement. It runs 4 iterations for the refinement. The model is trained on the TAP-Vid Panning MOVI-E dataset from the paper, using 24 frames, 256 points per frame, and including the same color augmentations.

For completeness we have also computed the offline default TAPIR results. Indeed, we find that online TAPIR works reasonably close to the offline variant (obtaining $0.825$ AJ on Davis-8 and $0.584$ AJ on Davis-full), likely because all query points are all from first frame.

**DPT**  To provide a reference point to standard architectures employed for monocular depth prediction we make use of the Dense Prediction Transformer (DPT) [47] architecture. We follow a similar set-up as in Ranftl et al. [47], where we configure DPT with four reassemble and fusion blocks. Each block processes a $16 \times 16$ feature map from a pre-trained ViT at multiple spatial resolutions $4 \times 4$, $8 \times 8$, $16 \times 16$ and $32 \times 32$. We use a feature dimensionality of 256 for each block and 128 dimensionality for the depth estimation head. Similar to in Dehghani et al. [11] we reuse the same ViT feature map at each stage. ViT features maps are obtained by upsampling the input frame to $224 \times 224$ for ViT-E and ViT-22b, which use a patch size of 14, and to $256 \times 256$ for ViT-L, which uses a patch size of 16. We use ReLU normalization on the output of DPT to ensure that it is non-negative, and adjust the value range by inverting the model prediction. The output resolution of DPT is at $224 \times 224$ and we upsample the depth targets and mask accordingly. DPT is trained and evaluated on randomly sampled video frames of Waymo using the same crop augmentations as mentioned in the previous section.

## C.4  Qualitative Experiments

For the qualitative experiments in Section 4 we trained a MooG model on a dataset mixture from Ego4D [21] and Kinetics700 [6]. The architecture used is identical to other models specified here. The model was train on 8 frame long sequences randomly subsampled from the longer sequences in the data. The model was trained with batch size of 256 from 1M steps and takes about 3 days on $8 \times 8$ TPU cores to train, though results are already quite good after a few hours of training.

To generate the attention maps in Figures 4 and 7 and the supplementary material, we first unroll the model on a test sequence to produce next frame predictions - this can be done on sequence lengths much longer than the training sequence length. For each frame, we take each decoded pixel coordinate (remember these are the queries that are used as input in the decoder) and average the attention for each token across all decoder layers. This tells us how much attention a specific token contributes when decoding a specific image location at a specific time step. We can visualize specific token attention maps through time as in Figure 4 or we can take the arg-max across all token for a specific location and colour code the tokens to get a more complete view of which token is most responsible for every image location across time 7 and videos in the supplementary material. Here the "grid-tokens" are obtained from the *Grid Rec.* baseline, trained in the same manner as MooG.

PCA visualizations (Figure 5 were generated by unrolling the model on a batch of 24 sequences, each 12 frames long. We take the set of predicted tokens and concatenate all of them across the first dimension, resulting in a $294,912 \times 512$ matrix. We then perform PCA analysis with this data. taking the leading 64 components, resulting in a $294,912 \times 64$ matrix which we reshape back to the original unrolled size up to the channel dimension to size $24 \times 12 \times 1024 \times 64$. In order to visualize these in image space we use the argmax token for each image location. We observe (manually) that the first 20 or so components correspond to the positional encoding and contain no or very little scene relevant content. Many of the components are informative, however — corresponding to different elements in the scene, or properties such as colour, motion etc. In order produce the visualization in Figure 5 we chose 3 of the "interesting" components and place them in RGB channels, visualizing the result as a video.

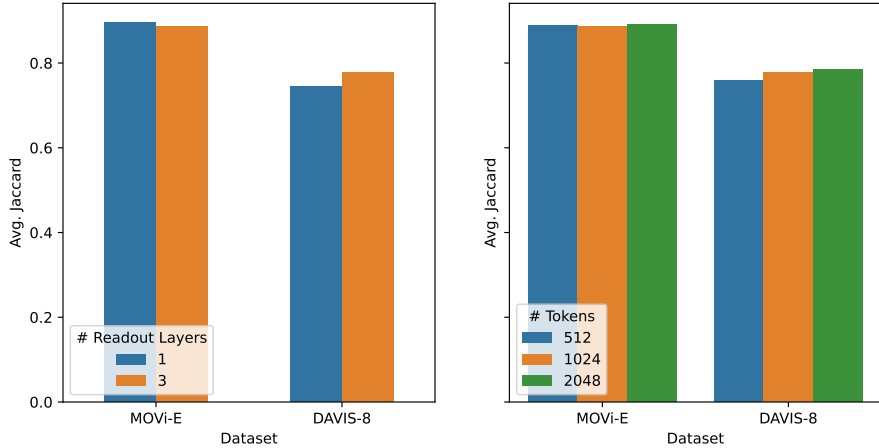

Figure 6: We vary several key hyper-parameters of MooG and report results for the end-to-end point tracking setup.

# D Additional Results

**Analysis** We report the results for our hyper-parameter study of MooG in Figure 6. Here MooG was trained end-to-end on MOVi-E together with the point tracking objective, identical to the setup for Table 2. We observe a marginal improvement using a single transformer layer in the point readout (Table 6) on MOVi-E, but a decrease on DAVIS-8. When changing the total number of tokens, we see a very slight improvement having additional tokens. We report qualitative results for changing the number of tokens at inference time in Figure 7.

We evaluated the effect of using different number of frames during training. We trained MooG end-to-end on MOVI with 2, 4 and 8 frames and evaluated on the point tracking and depth estimation downstream tasks as in Table 2. Here we observed that training on 8 frames is significantly better than training on 4 frames, which is in turn much better than training on 2 frames in the case of point tracking. In particular, on the DAVIS-full evaluation we obtain 0.51, 0.36, 0.16 AJ respectively. The differences between the three become more pronounced the longer the evaluation sequence is — presumably the model learns longer term dynamics when training on longer sequences, which improves its generalization ability to different sequence lengths. For depth evaluation we do not observe a major difference between 8 and 4 frames (but both are much better than 2 frames), obtaining AbsRel error of 0.03, 0.03 and 0.19 respectively on MOVi depth.

**Variance** Many of the results reported for MooG are for a single seed. To provide some indication of variance, we evaluated 3 seeds for the MooG variant reported in Table 3. We observe a standard error of the mean (SEM) of approx. 1% (absolute) for all reported metrics.

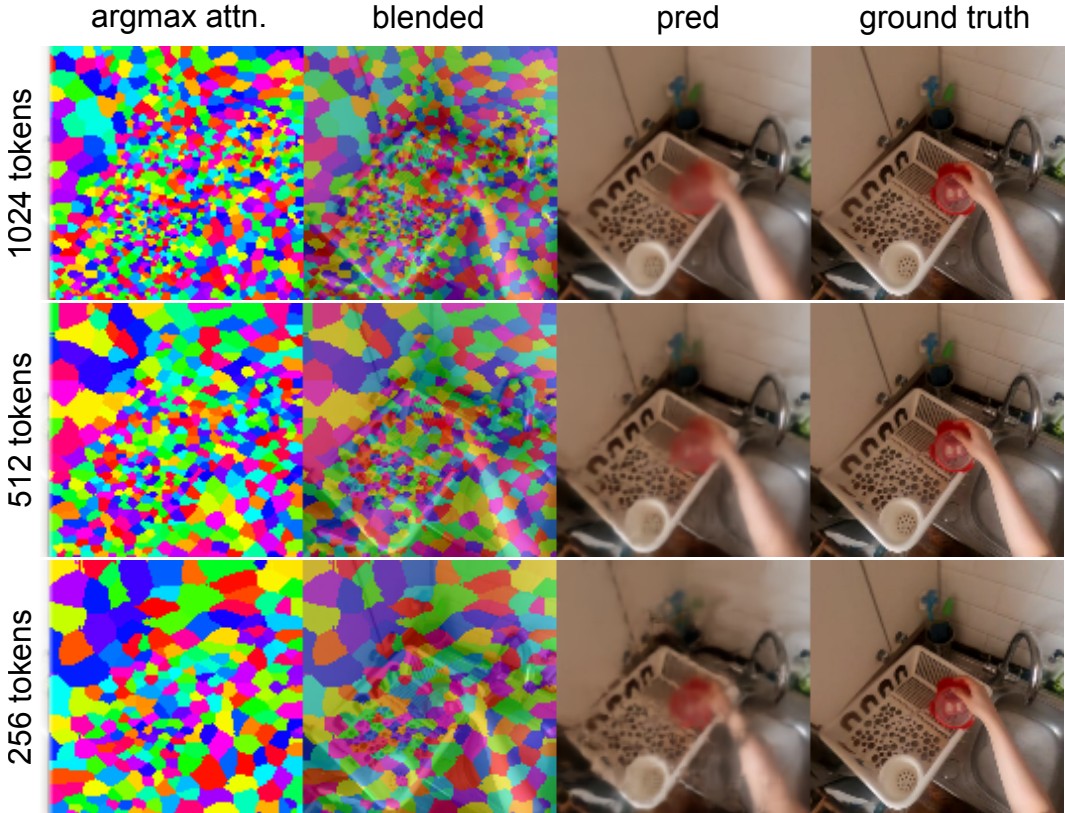

Figure 7: MooG can be instantiated with a different number of tokens at test time *without retraining*. Because the model architecture is independent of number of tokens and image resolution, we are free to choose the number of tokens used at evaluation time. The model depicted here was trained with 1024 tokens. We instantiate the model with 256, 512 and 1024 tokens. As can be seen the model has no problem in adapting to a different number of tokens, producing good predictions while tokens cover more area in the image as needed.

