# OpenReview forum: "Moving Off-the-Grid: Scene-Grounded Video Representations"
_NeurIPS.cc/2024/Conference — NeurIPS 2024 spotlight_

### Official Review · Reviewer_ykZT · 2024-07-09

**Soundness:** 3
**Presentation:** 3
**Contribution:** 3
**Rating:** 6
**Confidence:** 4

**Summary:**

This work presents a video representation learning approach where tokens are decoupled from explicit grid locations in the video sequence.  Rather than simply extract patches to construct tokens and apply self-attention across blocks of frames, this proposed model predicts sub-pixel motion given a history of frames to advect latent features throughout the video.  Once those grid-aligned features are advected to the next frame to their non-grid aligned positions, cross attention can be applied between the grid aligned observed features and the advected features in order to correct the result.

The authors claim three contributions:
* Introduction of MooG
* Qualitative validation of the approach
* Quantitative evaluation on a number of different downstream tasks

**Strengths:**

Idea seems like a fundamental one that the community will use.  It applies to a wide variety of tasks.  The concept is novel and general.  There's a lot of great qualitative experiments that help me understand that this method is implicitly learning a tracking representation.

**Weaknesses:**

I had a hard time understanding the point of the corrector for awhile.  If the predictor functions as intended, shouldn't a reconstruction loss applied to D(P(z)) be enough?  But after some more thought, I guess the intent isn't to build a an open loop video predictor (although I suppose the authors could evaluate it as such?), but instead the predictor is just a way to establish alignment between features in neighboring frames so that the cross attention of the corrector is more effective?

I think the prior work could go farther back in history in order to frame this proposed prediction-correction framework as being related to old school recursive filters.  For example, I think if I were told how this is designed similar to a Kalman filter, then I'd have a much clearer picture of what is trying to be done.

I don't think Fig 1 does enough to clarify the system.  I felt it was still necessary to read eqn 1 and 2 to understand what exactly was going on.  Some ideas include: (1) include arrows going up in the direction of image -> encode -> state and state -> decode -> image. (I originally read this as left-to-right top-to-bottom and realized I went the wrong direction once I saw encode and decode -- arrows would prevent this). (2) Make the relationship between prediction and correction clear.  It's currently presented as two colorful feature map sets, which doesn't communicate much.  Once again, if it's established early on what the predict-correct framework looks like, then this is obvious, but I failed to grok it on the first read through.

**Questions:**

I didn't get a good impression of what kind of motion dynamics the predictor properly supported.  It looked like there was some overshooting happening in the gifs, but it's not clear to me if this is a real issue.  Can this be clarified?

One ablation that would be useful is to change the backprop through time length.  It would be great if there was evidence that only 2 or 3 frames were enough to get the predictor to estimate dynamics.  Another question could be, what kind of dynamics are modeled with 2, 3, 4, 8, etc. frames and are there diminishing returns?  Can this be clarified?

Given that this model is deterministic and subject to over-blurring in the reconstruction task, have the authors explored other losses such as an LPIPS loss which might be more invariant to small misalignments?

**Limitations:**

looks good to me

---

> ### Author Rebuttal · Authors · 2024-08-06
>
> Thank you for your detailed review and constructive feedback. We were pleased to hear that you consider our contribution to be novel, general, and the ideas widely applicable. We appreciate that you recognize the strength of our qualitative experiments to help understand the method’s working.
>
> **“I had a hard time understanding the point of the corrector for a while. [...] the predictor is just a way to establish alignment between features in neighboring frames so that the cross attention of the corrector is more effective?”**
>
> Thank you for pointing out this confusion. We will clarify this in the updated version of the paper. Indeed, as is mentioned in the paper, the separation between the corrector and predictor is slightly artificial if the predictor is not unrolled in an open-loop prediction set-up. Since this is a deterministic model, unrolling in an open loop would produce blurry predictions quite quickly, but this indeed an interesting future direction.
>
> The purpose of the corrector is to integrate new information in the state based on the current observation. However, if we were to decode from this state it could create a shortcut that the model could exploit (the model could learn to ignore the temporal prediction component, i.e. the alignment between features in neighboring frames, and just learn to auto-encode via the corrector). This is why we decode from the predicted state, which forces temporal prediction. For readout of down-stream tasks, we additionally make use of the corrected state, which combines both the prediction and the current observation.
>
> **“I think the prior work could go farther back in history in order to frame this proposed prediction-correction framework as being related to old school recursive filters. For example, I think if I were told how this is designed similar to a Kalman filter, then I'd have a much clearer picture of what is trying to be done.”**
>
> Thank you for pointing out the connection to Kalman filters and recursive filters. Indeed, there are some resemblances worth pointing out that we could comment on in the related work section. We will revise the text accordingly, which hopefully improves the overall presentation.
>
> **“I don't think Fig 1 does enough to clarify the system.”**
>
> Thank you for pointing this out. We intend for the revised draft to make the connection between the corrector and the predictor clearer. Further, we have updated the main model figure with this in mind, which can be found in the supplementary pdf attached to the general response. Please let us know if you have any suggestions for further changes.
>
> **“I didn't get a good impression of what kind of motion dynamics the predictor properly supported. It looked like there was some overshooting happening in the gifs, but it's not clear to me if this is a real issue. Can this be clarified?”**
>
> What the predictor would learn exactly is difficult to quantify as it will depend on many factors - model capacity, training data and so on. We do observe that when there is fast motion the model tends to lose tracking and assigns a new token to the moving element. This makes sense considering that the amount of uncertainty grows with faster motion. Whether this is a real issue will similarly depend on the nature of the down-stream tasks. Evidently, for our current evaluation spanning several standard computer vision tasks, this is not a major issue. However, for more complicated dynamics the predictor could benefit from a more specialized architecture.
>
> **“One ablation that would be useful is to change the backprop through time length. It would be great if there was evidence that only 2 or 3 frames were enough to get the predictor to estimate dynamics. Another question could be, what kind of dynamics are modeled with 2, 3, 4, 8, etc. frames and are there diminishing returns? Can this be clarified?”**
>
> Thank you for this suggestion. Generally speaking, we’d anticipate that a minimum of 3 frames is needed to make good predictions as acceleration information requires 3 different measurements - with 2 frames only velocity information is available. We agree that it would be interesting to ablate this hyper-parameter and we will commit to doing so in the updated version of the paper.
>
> Previously we have explored training with a probabilistic “stop-gradient” that activated at random times through the sequence to effectively train on 3-4 frames long sequences but not overfit to a specific length. However as the model development continued, we no longer found this to be needed.
>
> **“Given that this model is deterministic and subject to over-blurring in the reconstruction task, have the authors explored other losses such as an LPIPS loss which might be more invariant to small misalignments?”**
>
> Thank you for this suggestion. In fact, we have not explored LPIPS or other perceptual losses to mitigate this, though this would be an interesting direction for future work. Integration of perceptual losses into MooG is not straightforward as we decode a random subset of pixels at every time step for efficiency reasons, as pixels directly serve as queries in the transformer decoder. Future work could explore decoding (latent) patches instead of raw pixels to mitigate this issue, which would make the model compatible with perceptual losses. A related direction that is worth pursuing is to have the state be the outcome of a sampling process (eg. as in a diffusion model or VAE), which might also help address blurry predictions.

---

> > ### Comment · Reviewer_ykZT · 2024-08-12
> > **Concerns addressed**
> >
> > My concerns were addressed in the rebuttal and I will keep the rating of weak accept.

---

### Official Review · Reviewer_RPGH · 2024-07-12

**Soundness:** 3
**Presentation:** 3
**Contribution:** 3
**Rating:** 6
**Confidence:** 3

**Summary:**

This work discusses Scene-Grounded Video Representations. Compared with current vision models that make each layer consist of tokens organized in a grid-like fashion, the authors introduce Moving Off-the-Grid (MooG), a self-supervised video representation model that proposes an alternative approach. The novelties are: Introducing Moving Off-the-Grid (MooG), a novel transformer-based recurrent video representation model designed to learn off-the-grid (OTG) representations using a straightforward next-frame prediction loss; illustrating how this representation enhances a range of downstream vision tasks, including point tracking, monocular depth estimation, and object tracking.

**Strengths:**

+ This work allows individual tokens to consistently track elements of the scene through videos of arbitrary length and “anticipate” where a scene element will be observed next.
+ The authors designed MooG to process an unlimited number of video frames while consistently maintaining a scene-grounded representation of the video.
+ Tokens within the latent state are inherently associated with particular pixel positions. The model can obtain corresponding token representations based on different features of the image.

**Weaknesses:**

- In line 135, you said “It is, however, crucial that the image is decoded only from the predicted state—decoding the current frame from the corrected state reduces the problem to simple auto-encoding and hurts representation quality considerably.” So how to prove the effectiveness of the  Corrector module?
- What’s the difference between MooG and Deformable DETR? Does the Corrector play a same role as the offset modal in Deformable DETR? Besides, do you treat the states as the tokens?
- It seems that you used transformer (attention module) in Corrector and Predictor, and how does the algorithm perform in tracking tasks? What is its speed? Can it meet real-time requirements? How’s the performance in the Multi-object tracking field?
- In table 3, the performance of TAPIR on Davis-full is better than MooG, so could you please provide more detailed explanations?

**Questions:**

Please refer to the Weaknesses

**Limitations:**

The authors do some analysis and the work will not have any potential negative societal impact.

---

> ### Author Rebuttal · Authors · 2024-08-06
>
> Thank you for your detailed review and constructive comments.
>
> **“how to prove the effectiveness of the Corrector module?”**
>
> Thank you for your comment. Note that the corrector is the only part of the model that has access to the observation at the current time-step, i.e. to “correct” the prediction, and hence is the only channel for integrating new information. However, it also introduces a shortcut that bypasses the previous state when using the corrected state for reconstructing the observation. This is what necessitates both a predicted state (to be used for reconstruction that requires temporal prediction) and a corrected state (to be used for readouts that incorporate the current observation and is maximally informative).
>
> **“What’s the difference between MooG and Deformable DETR? Does the Corrector play a same role as the offset modal in Deformable DETR? Besides, do you treat the states as the tokens?”**
>
> One of the key differences between MooG and Deformable DETR is that Deformable DETR utilizes sparse spatial sampling of deformable convolution on the 2D feature map (with explicit multi-scale features), while MooG models the image as an off-the-grid set of latent representations. There is no sparsity nor offset in the corrector in MooG, whose purpose is to “corrrect” the state predicted from the previous timestep based on the current observation. The corrector cross-attends to the feature maps and updates the set of latent features through transformer layers directly. MooG is easier to implement, it is mainly attention blocks, there is no need for any specialized kernel (while deformable DETR requires specialized [deform_attn_cuda.cu](https://github.com/fundamentalvision/Deformable-DETR/tree/main/models/ops/src/cuda)).
>
> **“It seems that you used transformer (attention module) in Corrector and Predictor, and how does the algorithm perform in tracking tasks? What is its speed? Can it meet real-time requirements?”**
>
> MooG is a very lightweight model, totalling approximately only 35M parameters during training (and fewer during inference). In particular, the transformers layers you mentionhave only 2 or 3 layers (predictor and corrector), while the conv-net is only 6 layers which should be no problem to run at real time on modern hardware.
>
> Further, for readouts (like point/box tracking) the pixel decoder is not needed at inference time. The model performs well even when reducing the number of available tokens considerably (as shown in Figure 6 in Appendix C), which further improves speed.
>
> **“In table 3, the performance of TAPIR on Davis-full is better than MooG, so could you please provide more detailed explanations?”**
>
> Thank you for pointing this out. Indeed, because of the auto-regressive nature of the readout module, error accumulates when unrolling readouts over long sequences. This is not an issue with the base representation (as the corrector receives continuous observations) but we have observed how the read-out module ends up drifting eventually, i.e. as the initial conditioning signal (box or point tracks in the first frame) becomes less informative.
>
> We note how TAPVid and TAPIR are domain-specific SOTA approaches that leverage specialized architectures, such as explicit cost volumes, to mitigate such issues for a particular domain. They further have access to both past as well as future states, while MooG is a causal model with access only to past frames, i.e. it can be used for online tracking. MooG learns representations that are useful for a variety of downstream tasks, and we have not incorporated domain-specific improvements in the readout decoder. One interesting direction for future work might be how to make MooG representations perform competitively on one single domain by incorporating more specialized components in the readout decoder. We will update the draft to give more context to this comparison and why we do not necessarily expect to beat domain-specific SOTA methods with MooG.

---

### Official Review · Reviewer_r1sm · 2024-07-12

**Soundness:** 2
**Presentation:** 2
**Contribution:** 3
**Rating:** 6
**Confidence:** 4

**Summary:**

The authors present a self-supervised video representation learning strategy. A grid-structure free feature extractor is trained using a next frame prediction objective. A corrector module extracts per-frame features. A predictor module predicts the next frame features. A decoder (with suitable grid free architecture) reconstructs frames. A sparse (for efficiency) reconstruction loss is applied on decoder output as the learning signal.

The method, MooG, appears to bind features to specific structures in the visual inputs and tracks these across frames. The authors present evaluations for tracking and depth estimation.

**Strengths:**

1. Interesting and novel idea for self-supervised representation learning from videos
2. Clear explanation of methodology and through details

**Weaknesses:**

1. **Evaluations:**
Learned representations are evaluated on a) niche tasks, b) with only image SSL baselines. Proposed method uses videos to learn unlike the baselines. Please compare a) on generic tasks (evaluate learned representations for classification, object detection, or segmentation) and b) compare against video SSL methods (on the identified tasks and generic ones).
In fact, in introduction, the authors argue that tasks like object detection and segmentation need the kind of off-the-grid architectures being proposed. Evaluation on such tasks will strengthen the authors case.

2. **Prior Video SSL works:**
It is unclear how these representations compare against video SSL methods [2,3,4,5]. Maybe try to apply some of these methods on the selected tasks to compare how MooG representations compare to them? Or evaluate MooG on tasks these are commonly used for.

3. **Related Works:**
Consider discussing [1] which explores a similar idea of next frame prediction in a latent space to learn good representations. Also [6] which learns local features using language which shows DINO like grouping / tracking behaviour.


[1] Sequential Modeling Enables Scalable Learning for Large Vision Models, CVPR 2024
[2] Time Does Tell: Self-Supervised Time-Tuning of Dense Image Representations, ICCV 2023
[3] Is ImageNet worth 1 video? Learning strong image encoders from 1 long unlabelled video, ICLR 2024
[4] Self-supervised Video Transformer, CVPR 2022
[5] VideoMAE: Masked Autoencoders are Data-Efficient Learners for Self-Supervised Video Pre-Training, NeurIPS 2022
[6] Perceptual Grouping in Contrastive Vision-Language Models, ICCV 2023

**Questions:**

See weaknesses

**Limitations:**

Ok

---

> ### Author Rebuttal · Authors · 2024-08-06
>
> Thank you for your detailed review and constructive comments. We are pleased to find that you consider MooG to be an interesting and novel idea for self-supervised representation learning from videos, and that the presentation was clear to you.
>
> **“Please compare a) on generic tasks (evaluate learned representations for classification, object detection, or segmentation)”**
>
> Thank you for suggesting this comparison. MooG was designed with spatio-temporal grounding in mind, without placing a focus on high-level semantic features, and we chose evaluation domains that would demonstrate this most clearly. Keeping this mind in, we chose point tracking, box tracking, and depth estimation as low-level tasks that require little semantic understanding of a scene, which aligns well with our goal of learning scene-grounded, temporally-consistent representations. In preliminary experiments we observed that features learned by MooG are not well-suited for semantic tasks such as action classification in combination with linear readout heads. This is potentially due to the local nature of the representation or the simplicity of the readout setup we tried. We are happy to include a discussion of these preliminary observations in the paper and make suggestions for improvements for future follow-up work.
>
> **“It is unclear how these representations compare against video SSL methods [2,3,4,5]. Maybe try to apply some of these methods on the selected tasks to compare how MooG representations compare to them?‘)”.**
>
> Thank you for pointing this out. We have now added a comparison to VideoMAE v2 in the supplementary pdf attached to the main response, which is a more recent version of [5].
>
> We consider representations from 3 [publicly available VideoMAE v2 checkpoints](https://github.com/OpenGVLab/VideoMAEv2/blob/master/docs/MODEL_ZOO.md): the ViT-small and ViT-base variants, which contain 22M and 83M parameters respectively, as well as a ViT-giant model (1B params). The smaller variants were obtained by distilling the predictions of the ViT-giant model. We note that MooG contains approximately 35M parameters, which includes the pixel decoder.
>
> There are other differences that further make it difficult to compare, for example MooG was pre-trained on MOVI-E, while the VideoMAE models were trained on a superset of all Kinetics videos. MooG was trained purely in a self-supervised manner, while the VideoMAE v2 teacher network was finetuned for action recognition, and the ViT-small and ViT-base models were initialized from finetuned networks too.
>
> Keeping these differences in mind, we observe how MooG performs significantly better than the ViT-small and ViT-base sized models that offer similar parameter counts. Compared to the teacher ViT-giant model that has 30x more parameters, MooG performs very well: it is better on MOVi points and Waymo boxes, but worse on MOVi depth and DAVIS points. Both methods perform the same on MOVi boxes. We are hopeful that scaling MooG to 1B+ parameters and larger scale pretraining may lead to further improvements, though we leave this for future work as mentioned in our limitations section.
>
> **“Related Works: Consider discussing [1] which explores a similar idea of next frame prediction in a latent space to learn good representations. Also [6] which learns local features using language which shows DINO like grouping / tracking behaviour.”**
>
> Thank you for pointing this out. We will make sure to contextualize our work further w.r.t. autoregressive next-frame prediction methods (e.g. regarding the reference you provided which uses a grid-based VQ-GAN representation [1]), and with perceptual grouping / object-centric methods in our related work section. In short: end-to-end object-centric grouping methods are indeed closely related (as alluded to in our related work section), but this line of work typically aims at grouping entire objects or well-defined parts into a single object – this includes methods such as GroupViT [7] and the paper you referred to [6]. For more general, fine-grained spatio-temporal tasks such as point tracking or monocular depth estimation, enforcing an object bottleneck (e.g. 1 token per object) may be too restrictive: we find that downstream task performance significantly increases by providing the model with more capacity in terms of number of “off-the-grid” vectors/tokens that it can use to encode the video.
>
>
> **References**
>
>
> [1] Bai et al., Sequential Modeling Enables Scalable Learning for Large Vision Models (CVPR 2024)
>
> [2] Salehi et al., Time Does Tell: Self-Supervised Time-Tuning of Dense Image Representations (ICCV 2023)
>
> [3] Venkataramanan et al., Is ImageNet worth 1 video? Learning strong image encoders from 1 long unlabelled video (ICLR 2024)
>
> [4] Ranasinghe et al., Self-supervised Video Transformer (CVPR 2022)
>
> [5] Tong et al., VideoMAE: Masked Autoencoders are Data-Efficient Learners for Self-Supervised Video Pre-Training (NeurIPS 2022)
>
> [6] Ranasinghe et al., Perceptual Grouping in Contrastive Vision-Language Models (ICCV 2023)
>
> [7] Xu et al., GroupViT: Semantic Segmentation Emerges from Text Supervision (CVPR 2022)

---

> > ### Comment · Reviewer_r1sm · 2024-08-12
> > **Most concerns addressed; update to weak accept**
> >
> > I thank the authors for the detailed rebuttal. The authors address most concerns here. I believe this paper will be valuable to the community, hence support accepting.
> >
> > Suggestions:
> > Please update the paper with the explanation on why action recognition benchmarks is unfair to MooG. Highly recommend adding at least one line to the intro to explain this (while highlighting its unique strengths). It is useful to highlight the kind of tasks these learned representations are ideal for alongside limitations. Also, please mention this in detail somewhere in the paper,
> >
> >
> > Note on comparison to video SSL methods:
> > MAE style pre-training is known to not emerge strong grouping / tracking behavior (i.e. what DINO has). Consider evaluating against Video-SSL works building off DINO (e.g. [4] or preferably a similar more recent work). Maybe in later work or as an addition to the appendix.

---

### Official Review · Reviewer_MZrN · 2024-07-13

**Soundness:** 3
**Presentation:** 4
**Contribution:** 3
**Rating:** 7
**Confidence:** 4

**Summary:**

This paper proposes a field-based method for video representation learning called MooG. Instead of propagating a discretized grid of features for every pixel location in the video, the method updates an arbitrary set of state tokens. These state tokens are used to parametrize a context dependent PerceiverIO-style “field”: a network that decodes the RGB values at a particular location (x, y) in the target frame conditioned on the “predicted” state tokens from previous frames and a position encoded (x, y) input. Since the state tokens parametrize a field, they are dubbed “off the grid”. The paper proposes to train such a model in a recurrent fashion using next frame prediction. Further, it describes methods to read-out the (potentially pre-trained) representations for downstream tasks. Results show qualitatively that the off-the-grid representations are “well behaved” through compelling visualizations and quantitatively work better than on-the-grid counterparts for a range of dense visual tasks, such as object tracking and depth prediction.

**Strengths:**

-	The idea of combining recurrent state token estimation together with an off-the-grid field-like representation that is repeatedly queried across time and trained with next frame prediction is interesting and novel.
-	The methods section is simple and intuitive and the writing is clear and concise. At important steps (such as 130 – 132) where there could be room for confusion (eg. decoding from predicted tokens vs correct tokens), the details are highlighted to elucidate the key elements that make the method work.
-	A wide range of experiments are shown for three important dense prediction tasks: monocular depth estimation, points tracking and boxes tracking. Baselines are chosen appropriately, representing widely used and ubiquitous recent methods in the respective fields. Quantitative experiments show that MooG is a powerful representation learning method, and that it outperforms baselines in the frozen setting, which should be the dominant paradigm to evaluate a video pre-training model.
-	The qualitative visualizations are impressive and creative. That tracking (in the token attention sense) and segmentation features (in the PCA sense) emerge from using the limited token and off-the-grid inductive bias are exciting results. The paper shows many examples of these results.

**Weaknesses:**

-	While appropriate baselines are chosen for all downstream evaluations, and comparison is made with on-the-grid representations that simply propagate features, baselines with architectures similar to MOOG have not been explored in a lot of detail. In the related work, the “off-the-grid” similarity between Perceiver IO and MooG is acknowledged, with the key difference being that MooG additionally implements recurrence of state tokens.  One baseline that would be a closest comparison would be to train a Perceiver IO like model, except with spatio-temporal queries: in particular, this would entail conditioning on time and position. This could be trained simply like an auto-encoder (essentially reducing to Perceiver IO for video), or with next frame prediction.
- Papers that use similar architecture but specialized for domain-specific video prediction tasks are missing from the related works. While MooG is a general purpose representation learning method, a thorough discussion of some domain specific models would be important to highlight key differences between those papers and the potential general purpose nature of MooG representations. Some of these papers are already cited (eg. CONDITIONAL OBJECT-CENTRIC LEARNING FROM VIDEO, Kipf et al. 2022), but are missing deeper discussion on fundamental differences to MooG.

**Questions:**

-	Is the paper the first to show token binding to task related activities emerging unsupervised in a field-based transformer?
-	Are the baselines chosen for point tracking (TAP-Net, TAPIR) and for depth prediction (DPT, etc) the state of the art results in the domains? (Given MooG is a general purpose video representation learning method, I agree it would not be fair to compete against the best task-specific engineered methods, but a discussion on limitation and deficit to state of art (if any) would be useful).

**Limitations:**

The limitations section is adequate and thorough (discussing disocclusions, occlusions, and alternate coarse / fine tasks possible on video), and future directions for improvement are also duly noted in Appendix A.

---

> ### Author Rebuttal · Authors · 2024-08-06
>
> Thank you for your detailed review and constructive comments. We are pleased to hear that you found MooG interesting and novel, and recognized its considerable improvement over baselines in the frozen setting. We are glad that you found the writing and visualizations of high quality.
>
> **“One baseline that would be a closest comparison would be to train a Perceiver IO like model, except with spatio-temporal queries”**
>
> Thank you for suggesting this baseline, which is indeed a very sensible one. In fact it is close to one variant of the model we experimented with in the early phases of the project. At that time, we experimented with a non-autoregressive version of our model that mapped all encoded frames in parallel to a joint set of latent tokens (a single set for multiple time steps) and finally replicated this set across time steps with added temporal position encoding for decoding.
>
> Such an approach is in fact very similar to Perceiver IO for video (with some small design differences in encoder, decoder and position encoding). In preliminary experiments, we found that when evaluated on 8-frame DAVIS point tracking, both models performed roughly equally well. However, we faced challenges continuing the point tracking predictions of the parallel (Perceiver IO-style) model when moving beyond the 8-frame window that it was trained on. On the contrary, this was straightforward to achieve with MooG due to its autoregressive formulation.
>
> **“Papers that use similar architecture but specialized for domain-specific video prediction tasks are missing from the related works”**
>
> Thank you for pointing this out. We will deepen the discussion of prior works in the related work section with regard to domain-specific approaches (such as SAVi), and highlight similarities and differences to MooG where relevant.
>
> **“Is the paper the first to show token binding to task related activities emerging unsupervised in a field-based transformer?”**
>
> To the best of our knowledge, this paper is among the first to *explicitly study* this relationship in standard transformer architectures trained in a self-supervised fashion. That said, it is difficult to make a precise claim about this due to numerous (domain-specific) prior works such as SAVi and PARTS exploring architectures that display similar behavior at a coarser scale (i.e. object-level binding) and are at least “transformer-like”. It is also worth mentioning the recent body of work on 4D Gaussian splatting (e.g. Luiten et al., Dynamic 3D Gaussians, 2023), which aims to learn explicit 3D Gaussian representations that track elements of a dynamic scene, typically by using more explicit geometric constraints instead of generic transformer-based architectures.
>
> **“Are the baselines chosen for point tracking (TAP-Net, TAPIR) and for depth prediction (DPT, etc) the state of the art results in the domains? […] a discussion on limitation and deficit to state of art (if any) would be useful).”**
>
> TAPIR is very close to state of the art in the point tracking domain, and only outperformed by the very recent BootsTAP [1], which extends TAPIR by augmenting the pretraining dataset with unlabeled YouTube videos. The DPT architecture itself is commonly used in competitive monocular depth-estimation approaches such as the MiDaS line of work [2]. However, to achieve competitive performance, such approaches are usually pre-trained on multiple different depth estimation datasets and adopt more sophisticated losses to handle depth targets from varying sources in a single model. More recently, diffusion-based approaches have shown great promise for monocular depth estimation [3, 4], which move away from the DPT architecture.
>
> We agree that it would be useful to contextualize these baselines a bit better relative to SOTA in the field, and we will update the paper accordingly.
>
> [1] BootsTAP: Bootstrapped Training for Tracking-Any-Point
> [2] MiDaS v3.1 -- A Model Zoo for Robust Monocular Relative Depth Estimation
> [3] Repurposing Diffusion-Based Image Generators for Monocular Depth Estimation
> [4] Depth Anything: Unleashing the Power of Large-Scale Unlabeled Data

---

> > ### Comment · Reviewer_MZrN · 2024-08-11
> >
> > Thank you for the rebuttal. My questions and concerns have been adequately addressed, and I will maintain my score and recommendation for acceptance.

---

### Author Rebuttal · Authors · 2024-08-06

We thank the reviewers for their helpful comments.We are pleased to hear that the reviewers find our paper to be well-presented (MZrN13, RPGH11, ykZT08), interesting (MZrN13, r1sm12), and novel (MZrN13, r1sm12, ykZT08). Reviewers MZrN13 and ykZT08 positively highlight the quality of the experimental evaluation. Reviewer ykZT08 further notes that the “idea seems like a fundamental one that the community will use”.

We address the concerns raised by each reviewer individually. Further, the supplementary pdf includes several additional baselines and updated results:

* We have included a comparison to DINOv1 and DINOv2 in the end-to-end setting, which was previously missing. It can be seen how both approaches perform considerably better to their frozen counter-parts, yet MooG performs considerably better in both settings.

* After submission, we discovered a bug in the “Grid Rec.” baseline where no gradients were being back propagated into the encoder. After re-running, we now observe how “Grid Rec.” consistently outperforms the “Grid” baseline without recurrence, as one might expect. Compared to MooG, in the frozen setting (which is the main paradigm of interest), “Grid Rec” continues to perform considerably worse than MooG. In the end-to-end setting, where we finetune the model on the down-stream task of interest, we now observe how “Grid Rec” and MooG perform comparably, suggesting that the added supervision can help balance out architectural differences.

* Based on reviewer r1sm12’s suggestions, we have included a comparison to VideoMAE v2 [1]. We consider representations from 3 [publicly available VideoMAE v2 checkpoints](https://github.com/OpenGVLab/VideoMAEv2/blob/master/docs/MODEL_ZOO.md): the ViT-small and ViT-base variants, which contain 22M and 83M parameters respectively, as well as a ViT-giant model (1B params). The smaller variants were obtained by distilling the predictions of the ViT-giant model. We note that MooG contains approximately 35M parameters, which includes the pixel decoder. There are other differences that further make it difficult to compare, for example MooG was pre-trained on MOVI-E, while the VideoMAE models were pretrained on a superset of all Kinetics videos. MooG was trained purely in a self-supervised manner, while the VideoMAE v2 teacher network was finetuned for action recognition, and the ViT-small and ViT-base models were initialized from finetuned networks as well. Keeping these differences in mind, we observe how MooG performs significantly better than the ViT-small and ViT-base sized models that offer similar parameter counts. Compared to the teacher ViT-giant model that has 30x more parameters, MooG performs very well: it is better on MOVi points and Waymo boxes, but worse on MOVi depth and DAVIS points. Both methods perform the same on MOVi boxes. We are hopeful that scaling MooG to 1B+ parameters and larger scale pretraining may lead to further improvements, though we leave this for future work as mentioned in our limitations section.

* Based on reviewers’ ykZT suggestion we have revised the main model figure, a draft of which is included. We will continue to make revisions based on reviewer input and any additional feedback is much appreciated.


[1] VideoMAE V2: Scaling Video Masked Autoencoders with Dual Masking

---

### Decision · Program_Chairs · 2024-09-25

**Decision:**

Accept (spotlight)

**Comment:**

This paper received consistently positive final ratings of Accept, Weak Accept, Weak Accept, and Weak Accept. The reviewers felt that this work introduced a clear and novel idea that will likely see significant use in the field. It addresses a clear shortcoming with previous video representation learning methods. There was some discussion about an error with the reported performance metrics ("After submission, we discovered a bug in the “Grid Rec.” baseline") but the reviewers felt that this wasn't a major concern given the strong performance for the frozen-features case. Congratulations! Please take the reviewers' thoughtful comments into consideration as you prepare the camera-ready copy.